# The Brain's Bitter Lesson:
# Scaling Speech Decoding With Self-Supervised Learning

Dulhan Jayalath [1]   Gilad Landau [1]   Brendan Shillingford [2]   Mark Woolrich[3]   Oiwi Parker Jones [1]

## Abstract

The past few years have seen remarkable progress in the decoding of speech from brain activity, primarily driven by large single-subject datasets. However, due to individual variation, such as anatomy, and differences in task design and scanning hardware, leveraging data across subjects and datasets remains challenging. In turn, the field has not benefited from the growing number of open neural data repositories to exploit large-scale deep learning. To address this, we develop neuroscience-informed self-supervised objectives, together with an architecture, for learning from heterogeneous brain recordings. Scaling to nearly **400 hours** of MEG data and **900 subjects**, our approach shows generalisation across participants, datasets, tasks, and even to *novel* subjects. It achieves **improvements of 15-27%** over state-of-the-art models and **matches *surgical* decoding performance with *non-invasive* data**. These advances unlock the potential for scaling speech decoding models beyond the current frontier.

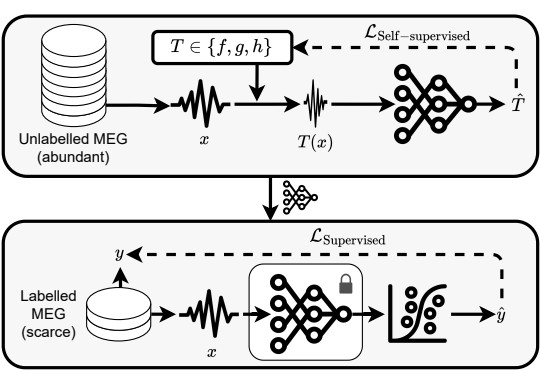

Figure 1: **Leveraging unlabelled data using pretext tasks for speech decoding.** We pre-train a neural network using tasks that generate implicit labels from abundant unlabelled MEG neuroimaging data, permitting learning from large heterogeneous datasets. The tasks apply a randomly selected neuroscientifically relevant transformation $T$ to the data and the network predicts the transformation. We then train a linear probe on top of the pre-trained model, which remains frozen, with labelled data, achieving superior generalisation owing to the strength of the representation.

## 1. Introduction

In his *Bitter Lesson*, Richard Sutton argues that a major conclusion of 70 years of AI research is that general methods exploiting large-scale computation will outperform model-based approaches as the availability of compute increases (Sutton, 2019). The ability of deep learning to learn from ever-larger datasets has enabled seemingly arbitrary scaling with computation, leading to astounding advances across a diverse set of domains (Jumper et al., 2021; Caron et al., 2021; OpenAI, 2023; Radford et al., 2023).

In the domain of brain data, and of tasks like speech decoding, the bitter lesson has not yet been fully assimilated.

Recent work towards *brain-computer interfaces (BCIs)* have tried to scale up labelled datasets for individual subjects, using either invasive (Moses et al., 2021; Willett et al., 2023) or non-invasive brain recordings (Tang et al., 2023), mapping these to transcripts of attempted or imagined speech. Yet, a number of obstacles to scale remain, especially in *magnetoencephalography (MEG)* data. Current speech decoding models rarely train on multiple subjects, combine datasets, or utilise data from diverse tasks. Thus the size of training data has been limited to how much can be acquired for a single subject, and data from other subjects, or from the growing number of public data repositories, has not been leveraged. There are many reasons for these limitations; individual brains and data from different neuroimaging scanners differ, for example. But overcoming these limitations holds the promise of training models on collective, internet-scale data.

[1]PNPL🍍 / [3]OHBA, University of Oxford [2]Google DeepMind. Correspondence to: <{dulhan, oiwi}@robots.ox.ac.uk>.

*Proceedings of the 42nd International Conference on Machine Learning*, Vancouver, Canada. PMLR 267, 2025. Copyright 2025 by the author(s).

https://pnpl.robots.ox.ac.uk/bbl

Decoding methods for MEG need to be highly data-efficient. While electroencephalography (EEG) data are abundant, MEG provides richer signals for decoding (Lopes da Silva, 2013; Hall et al., 2014) but are comparatively rare. Given the scarcity of speech-labelled MEG and the larger proportion of other MEG data, *self-supervised learning (SSL)* appears promising as it is an avenue for domains where labels are rare or hard to obtain (Balestriero et al., 2023). To data-efficiently learn from unlabelled MEG, we propose *pretext* training with neuroscience-informed input transformations that benefit downstream tasks. We use this for learning from unlabelled brain data (Figure 1) through an architecture for processing continuous multi-sensor neuroimaging signals. Our method provides a unified approach that enables leveraging data from other experiments that do not have the same labels (by treating them as unlabelled) and that come from different subjects and neuroimaging scanners. We evaluate representations learned with our approach on heard speech datasets acquired with MEG, setting the baselines for speech detection and voicing classification on this data.

Our main contributions are:

- A domain-specific **self-supervision method** and a **neural architecture** for representation learning from MEG that unlock scaling speech decoding over multiple subjects, multiple studies, and unlabelled data;

- Achieving **15-27% gains** over state-of-the-art self-supervised models, **matching *surgical* self-supervised decoding *non-invasively***, and showing **novel subject generalisation** for the first time in MEG; and

- Demonstrating evidence for **scaling laws** arising from pre-training with unlabelled MEG recordings using multiple times the volume of data in prior work.

## 2. Related Work

Prior work in speech decoding has focused almost entirely on supervised learning with decoding models that typically do not generalise across participants or experiments. This is true both in recent state-of-the-art invasive studies (Moses et al., 2021; Metzger et al., 2023; Willett et al., 2023; Chen et al., 2024a) and non-invasive studies (Tang et al., 2023). These prior works have scaled up the experimental data collected within individual subjects, but are unable to leverage data from other subjects and experiments. Nevertheless, the method developed by Tang et al. (2023) is remarkable for showing an ability to generalise across labelled task data. They do not, however, use unlabelled data or show cross-subject generalisation.

Specific studies into the limitations of generalising models between subjects show that while performance decreases on average when subjects are pooled, there are exceptions

(e.g. Anumanchipalli et al. (2019) and Makin et al. (2019) in surgical settings and Csaky et al. (2022) non-invasively). Défossez et al. (2023) show cross-subject generalisation for a segment identification task from participants listening to connected speech. However, they do not demonstrate generalisation to novel subjects and retrain their model for new datasets rather than being able to generalise across datasets or pool them. Their method is also unable to incorporate data without corresponding audio labels and so does not scale with other kinds of tasks.

In general, speech decoding has centred on different kinds of speech: listening, imagining, speaking out loud, and, for paralysed patients, attempting to speak aloud. We focus on listening because it is easier to decode than imagined speech (e.g. Martin et al. (2014)). There is also evidence of functional overlap between listening and imagined speech representations in the brain (Wandelt et al., 2024), though the question of overlap has been contested (Langland-Hassan & Vicente, 2018). While some work on decoding text directly from heard speech tasks with MEG and EEG exist, it is unclear whether these methods perform any better than a baseline that provides pure noise inputs to the model (Jo et al., 2024). Non-invasive speech decoding remains a highly challenging and unsolved domain.

Self-supervised pretext tasks have been successful in computer vision (Agrawal et al., 2015; Doersch et al., 2015; Noroozi & Favaro, 2016; Larsson et al., 2016; Zhang et al., 2016; Gidaris et al., 2018) but rarely applied to brain decoding. There are, however, methods that leverage unlabelled brain data in other ways (Banville et al., 2019; Kostas et al., 2021; Le & Shlizerman, 2022; Zhang et al., 2023; Yi et al., 2023; Cai et al., 2023; Ye et al., 2023; Yuan et al., 2024; Chen et al., 2024b). Unfortunately, most of this literature is unable to scale and harmonize heterogeneous non-invasive data. The most notable of these works include Wang et al. (2023), who learn contextualised embeddings of time-frequency input representations through masked spectrogram in-filling. Their impressive speech detection results were achieved with invasive neural recordings, which are comparatively rare and thus have less potential to scale than non-invasive data. Another, BIOT (Yang et al., 2023), learns from generic heterogeneous bio-signals with a contrastive pre-training objective, rather than masking, and applies it to ECG/EEG data. Notably, none of these methods optimise their objectives for speech decoding or focus on MEG.

## 3. Method

We introduce a neural architecture to embed heterogeneous brain signals. Then, we leverage this architecture for self-supervised learning from unlabelled MEG data using a set of pretext tasks designed to generate generalisable brain representations for speech decoding.

## 3.1. Network Architecture

Our neural network architecture has two stages (Figure 2): pre-training with pretext tasks on unlabelled data, and training a linear probe with labelled data for downstream tasks.

We divide recordings into windows of length $w$ seconds or $t$ samples. At train time, each batch of windows is standardised such that each sensor has zero mean and unit variance. The network takes as input the standardised sample windows. To combine heterogeneous datasets, which have varying numbers of sensors $S$, we apply a dataset-conditional linear layer to the sensor dimension, projecting the signal into a shared space with dimension $d_{\mathrm{shared}}$. Then, to encode the signal, we construct a wave-to-wave convolutional encoder architecture, the *cortex encoder*, inspired by work in neural audio codecs (Zeghidour et al., 2022; Défossez et al., 2022). Specifically, our convolutional encoder adapts the SEANet architecture (Tagliasacchi et al., 2020) used in Défossez et al. (2022) which we describe here and as part of Figure 2. As these codecs typically operate on mono audio signals in $\mathbb{R}^{1 \times t}$, while our signals are in $\mathbb{R}^{d_{\mathrm{shared}} \times t}$, we increase the convolutional channel dimension from 1 to match $d_{\mathrm{shared}}$ while also inflating the channel dimension of subsequent convolutions. We refer to the output dimension of embeddings from this backbone as $d_{\mathrm{backbone}}$. Thus, the backbone takes as input a window in $\mathbb{R}^{S \times t}$, and encodes this into $\tau$ embeddings, each of dimension $d_{\mathrm{backbone}}$ (i.e. an $\mathbb{R}^{d_{\mathrm{backbone}} \times \tau}$ output).

In the speech recognition literature, models include speaker conditioning to account for vocal and prosodic differences (Gibiansky et al., 2017). Just as speakers have different voices, neural responses between subjects have different characteristics. Consequently, individual variation leads to models that do not generalise well across subjects (Csaky et al., 2022). We address this with a similar approach to the speech literature by introducing subject conditioning using *feature-wise linear modulation (FiLM)* (Perez et al., 2018). As Zeghidour et al. (2022) find that conditioning is as equally effective at the encoder bottleneck as in other stages of the model, we also condition at the bottleneck.

Following Balestriero et al. (2023, Section 3.2), we use a two-layer projector to alleviate misalignment between our pretext and downstream tasks in the representation. After the projector, linear classifiers make predictions for each of the pretext tasks. When fine-tuning, we train a linear decoder, for a downstream task, on top of the pre-trained representation, which remains frozen. Thus, we backpropagate only through the classifier. A trainable dataset-specific linear layer can be introduced for a novel dataset.

For speech detection, our classifier makes a prediction for each individual embedding. For voicing classification, where there is only one label for each sample window, the

Table 1: **Functional frequency bands in brain activity.**

| Band | Hz | Association |
|---|---|---|
| Delta ($\delta$) | .1-4 | Rhythmic structure of heard speech (Luo et al., 2010) |
| Theta ($\theta$) | 4-8 | Tracking (Luo & Poeppel, 2007) and phase-locking to the amplitude envelope of heard sentences (Peelle et al., 2012) |
| Alpha ($\alpha$) | 8-12 | Attentional processes and the inhibition of irrelevant information (Strauß et al., 2015) |
| Beta ($\beta$) | 12-30 | Top-down predictive coding (Bressler & Richter, 2015) which affects lexical processing (Weiss & Mueller, 2012) |
| Gamma ($\gamma$) | 30-70 | Higher cognitive functions (e.g. memory, learning, reasoning, and planning) (Fries, 2009; Buzsáki & Wang, 2012) |
| High Gamma ($\gamma^{\mathrm{high}}$) | 70+ | Speech detection (Hamilton et al., 2018) and phonemic feature classification in the STG (Mesgarani et al., 2014) and the *ventral sensorimotor cortex (vSMC)* (Cheung et al., 2016) |

embeddings are flattened into a tensor in $\mathbb{R}^{d_{\mathrm{backbone}} \times \tau}$ representing the entire window. This is the input to the voicing classifier and is referred to as full epoch decoding in neuroimaging literature (Csaky et al., 2023).

### 3.2. Pretext Tasks

To use our architecture for pre-training, we construct pretext objectives for unsupervised learning of generalisable speech decoding features. These objectives are inherently agnostic to the sensor count because they operate on properties that are independent of the specific sensor arrangement. This key design choice enables the tasks to work seamlessly across datasets with varying numbers of sensors—a critical requirement for combining heterogeneous brain data.

**Band prediction.** In the literature, neural responses can be segmented into functional frequency bands (Giraud & Poeppel, 2012; Piai et al., 2014; Mai et al., 2016) (Table 1). Sensitivity to these frequencies would bring about functional separability in the representation space. Thus, to learn such representations, we train the network to classify rejected bands. As High Gamma is a relatively wide band we split it into two sub-bands: *Lower High Gamma* ($\gamma^{\mathrm{high}}_{\mathrm{lower}}$) waves (70–100 Hz) and *Upper High Gamma* ($\gamma^{\mathrm{high}}_{\mathrm{upper}}$) waves (100–150 Hz). Our task applies a band-stop filter for a randomly selected band $\omega$ to the sample $x$, passes the filtered sample

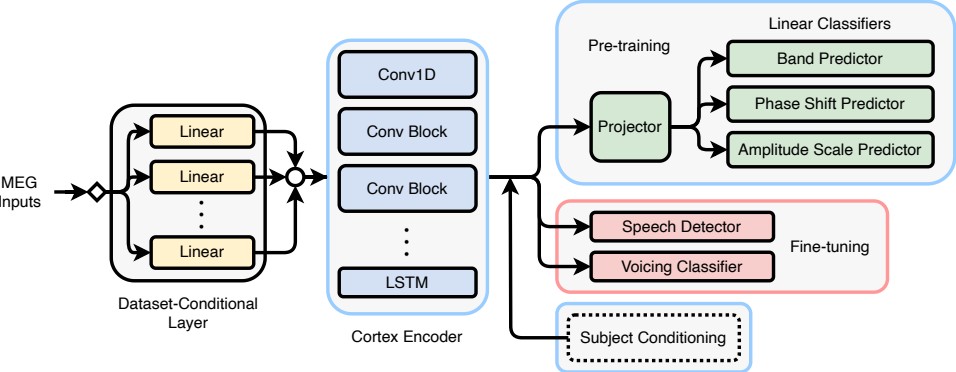

Figure 2: **Architecture overview.** Inputs are projected into a shared dimension by the dataset-conditional layer, then encoded. In pre-training, all weights are trainable except for modules in light-red, while in fine-tuning, modules with light-blue borders are frozen and modules with light-red borders are unfrozen. Dashed borders indicate optional components.

$x^{\omega'}$ through the network backbone $g$ and the corresponding linear predictor $f_{\text{band}}$, requiring the network to classify which frequency band $\omega$ was rejected. This yields the loss

$$\mathcal{L}_{\text{band}} = \sum_{x \in B} \mathcal{L}_{\text{CE}}(f_{\text{band}}(g(x^{\omega'})), \omega), \quad (1)$$

where $B$ is a mini-batch of samples, $\omega \in \{\delta, \theta, \alpha, \beta, \gamma, \gamma_{\text{lower}}^{\text{high}}, \gamma_{\text{upper}}^{\text{high}}\}$, and $\mathcal{L}_{\text{CE}}$ is the cross-entropy loss as this is a multi-class classification task.

**Phase shift prediction.** Phase coupling between networks of neuron populations is necessary for coordinating brain activity (Fries, 2005; Vidaurre et al., 2018) and so phase coupling between spatially distant sensors is likely to be a useful feature. Supporting this insight, recent work (Jiang et al., 2024) also finds phase to be an essential component of the signal.

To learn representations that encode phase differences between brain areas, this task applies a discrete uniform random phase shift $\phi \in \{0, \frac{\pi}{8}, \frac{\pi}{4}, \frac{3\pi}{8}, \frac{\pi}{2}, \frac{5\pi}{8}, \frac{3\pi}{4}, \frac{7\pi}{8}\}$ to a uniformly randomly selected proportion $\rho \in [0, 0.5]$ of the sensors. Applying this shift to random sensors is critical since sensors are placed in different positions, capturing different regions of the brain. Uniform random selection ensures differences between any two regions of the brain are represented. The objective of this task is to predict the phase shift. This leads to a similar loss

$$\mathcal{L}_{\text{phase}} = \sum_{x \in B} \mathcal{L}_{\text{CE}}(f_{\text{phase}}(g(x^{\phi})), \phi), \quad (2)$$

where $x^{\phi}$ describes the signal with a phase shift $\phi$ applied to a proportion of the sensors. We use a discrete number of possible phase shifts, treating it as a multi-class task rather than a regression task, to ease the difficulty of the problem as MEG scanners typically have a large number of sensors.

**Amplitude scale prediction.** MEG and EEG signals use an array of sensors at different spatial locations, capturing different signal sources more intensely. Representing the relative amplitude difference between sensors could be important for differentiating between neural responses originating from distinct parts of the brain. Within speech, Hamilton et al. (2018) find that localised regions of the STG respond to sustained speech and speech onsets. Differentiating between neural responses from this region and others may be essential for decoding speech perception.

Thus, this pretext task focuses on learning representations that encode relative sensor amplitude differences. Similar to the phase shift task, we select a random proportion of the sensors $\rho \in [0, 0.5]$ and apply a discrete random amplitude scaling coefficient $A \in [-2, 2]$, discretised into 16 scaling factors, to the signal. The objective is to predict the scaling factor, leading to the loss

$$\mathcal{L}_{\text{amplitude}} = \sum_{x \in B} \mathcal{L}_{\text{CE}}(f_{\text{amplitude}}(g(x^A)), A), \quad (3)$$

where $x^A$ is the signal scaled with $A$.

**Combined tasks.** These pretext tasks capture complementary time- and frequency-domain properties of the signal. Hence, during pre-training, we combine them, creating an augmented version of the input for *every* pretext task by applying the matching transformation. We feed the augmented inputs through the network backbone and apply the corresponding classifier to predict the transformation, summing the weighted losses such that our final pre-training loss is

$$\mathcal{L}_{\text{SSL}} = w_1 \mathcal{L}_{\text{band}} + w_2 \mathcal{L}_{\text{phase}} + w_3 \mathcal{L}_{\text{amplitude}}, \quad (4)$$

where $w_i$ is a constant coefficient for each self-supervised loss.

# 4. Experiments

We evaluate our self-supervised representations by measuring how they scale with unlabelled data and generalise across datasets, subjects, and tasks. We focus our evaluation on MEG data as the signal is rich, with better spatial resolution than EEG (Lopes da Silva, 2013) and faster sampling rates than fMRI (Hall et al., 2014). We pre-train all models to completion and then train a linear probe on labelled data for each task. In all tables and figures, we quote the *receiver operating characteristic area under the curve (ROC AUC)* where chance is always 0.5 regardless of the class distribution. We show the test ROC AUC at the best validation ROC AUC (early stopping) and quote uncertainty as the standard error of the mean over up to five seeds.

## 4.1. Experimental setup

**Datasets.** In total, we use almost five times the volume of data in prior MEG work, totalling approximately 400 hours with nearly 900 subjects across pre-training and downstream training. Unless specified otherwise, we pre-train with Cam-CAN (Shafto et al., 2014; Taylor et al., 2017) as an unlabelled representation learning dataset. This is a study containing 641 subjects with resting and sensorimotor tasks, totalling approximately 160 hours of MEG recordings. When aggregating datasets, we also pre-train with MOUS (Schoffelen et al., 2019), which contains 204 subjects and another 160 hours from visual and auditory tasks. Downstream, we use labelled heard speech MEG datasets where participants listen to short stories or audiobooks. We mainly focus on Armeni et al. (2022) which contains 3 subjects who listen to 10 hours of recordings each (30 hours total). We also analyse Gwilliams et al. (2023) which has 27 subjects, each recorded for 2 hours (54 hours total). The latter dataset is particularly difficult to decode from as there is very little within-subject data and it did not enforce the use of head casts to immobilise participants. Nevertheless, given it has many more subjects, we use this dataset to study subject generalisation.

**Preprocessing.** Each recording is in $\mathbb{R}^{S \times T}$ where $S$ is the number of sensors and $T$ is the number of time points sampled by the scanner. To eliminate high-frequency artifacts, we apply a low-pass filter at 125Hz as well as a high-pass filter at 0.1Hz to remove slow-drift artifacts. Since the datasets were recorded in Europe, where the electric grid frequency is 50Hz, we apply a notch filter at 50Hz and its harmonics to account for line noise. Treating the low-pass filter threshold as the Nyquist frequency, we downsample the signal to twice that at 250Hz, avoiding aliasing within our band of interest. Finally, we detect sensor channels with significant noise and artifacts using a variance threshold and replace them by interpolating the spatially nearest sensors.

**Downstream tasks.** We evaluate our methods primarily on *speech detection*. This task determines whether speech occurs in the auditory stimulus using the neural response. This is a fundamental task in understanding speech perception and is one of the few tasks that so far show statistically significant results in highly noisy MEG signals. It also has direct applications to BCIs as it can be used to segment words or sentences for decoding and activate a speech BCI when a patient wishes to communicate. Secondarily, we also study *voicing classification* to demonstrate the versatility of our representations for general speech decoding tasks. Given data aligned at the onset of a phoneme, the task is to recognise whether the phoneme is *voiced* or *voiceless*, where voicing is a phonetic feature that categorises whether a speech sound is associated with vocal cord vibration. This task is also directly relevant to a speech BCI as it involves classifying phonemes which can be used to decode words.

## 4.2. Learning Generalisable Representations Using Pretext Tasks

Table 2 shows that our approach achieves two key feats: outperforming comparable state-of-the-art self-supervised methods by 15-27% (part C), and matching the performance of prior self-supervised methods with surgical data (11) while using only non-invasive data. Since non-invasive data has a much lower signal-to-noise ratio than surgical data, this is quite unprecedented. In the rest of this section, we analyse this table in detail.

In part B, we show the results of pre-training models with each pretext task independently, together, and without any pre-training at all. Using pretext tasks (3, 4, 5) outperforms no pre-training (2). Interestingly, the combination of all pretext tasks (5) leads to better generalisation than any task on its own (the improvement over (3) is statistically significant). We conjecture that this is because our pretext tasks capture complementary properties in time- and frequency-space, ensuring that the representation includes more salient features than any individual task could encode. Finally, we apply Gaussian filtering to the predictions (7), smoothing out anomalies in the predicted speech envelope.

Next, we turn to the baselines, starting with Table 2 part A. Our method outperforms random selection (0) and training a linear layer with the MEG signal directly (1). The latter even has substantially more trainable parameters because the input dimension is larger without an encoder. In part C, we compare our approach to two state-of-the-art self-supervised methods. In each experiment, we apply Gaussian filtering as in (7). Although better than random, BrainBERT (9) does not generalise as well as our method (11). BrainBERT employs a generic masked spectrogram in-filling pre-training objective. While it is intended for speech decoding tasks, the pre-training objective is not designed to specifically capture features salient to neural speech processing. Furthermore,

Table 2: **Our approach surpasses baselines in speech detection by up to 27% and matches surgical decoding.** For *linear*, we train a supervised linear classifier on the MEG signals. For ours and BrainBERT, we train a linear layer on top of a backbone pre-trained on CamCAN, with the rest of the model frozen. For BIOT, we use their pre-trained weights. In the *no pre-training* baseline, the backbone uses randomly initialised and frozen weights. In the surgical context, we quote the result from Wang et al. (2023, Table 2). With *all* pretext tasks, losses are weighted equally.

| Part / ID | | Model | | ROC AUC |
|---|---|---|---|---|
| A | 0 | Random select | | .500 |
| | 1 | Linear | | $.539_{\pm.002}$ |
| B | 2 | **Ours** | No pre-train. | $.519_{\pm.002}$ |
| | 3 | + linear | $\text{Amp}_{(\rho = 0.2)}$ | $.624_{\pm.001}$ |
| | 4 | | $\text{Phase}_{(\rho = 0.5)}$ | $.615_{\pm.001}$ |
| | 5 | | Band | $.588_{\pm.001}$ |
| | 6 | | All tasks | $.630_{\pm.000}$ |
| | 7 | | + smoothing | $\mathbf{.700}_{\pm.002}$ |
| C* | 8 | BIOT[1] + linear | | $.615_{\pm.002}$ |
| | 9 | BrainBERT[2] + linear | | $.556_{\pm.007}$ |
| | 10 | EEGPT[3] + linear | | $.602_{\pm.006}$ |
| | 11 | **Ours** (best) + linear | | $\mathbf{.705}_{\pm.003}$ |
| | 12 | BrainBERT[2] + lin. (surgical) | | $.71_{\pm.06}$ |

[1]Yang et al. (2023) [2]Wang et al. (2023) [3]Wang et al. (2024)

their method is less data-efficient because in-filling is a harder generative task compared to classification and while their methodology was developed for relatively high-fidelity intracranial recordings, the inherently lower signal-to-noise ratio of MEG presents an even greater challenge.

Our last baselines are BIOT (8) and EEGPT (10). We leverage publicly released weights pre-trained with thousands of hours of EEG. However, both still fall short of our pre-training method for reasons which we believe are similar to BrainBERT—their objectives do not leverage neuroscientific understanding of speech processing.

Finally, and quite remarkably, our best result matches the AUROC quoted in Wang et al. (2023, Table 2) who use *intracranial* data from heard speech (12). We achieved this score with *non-invasive* data which is typically substantially more difficult to decode due to the low signal-to-noise ratio.

### 4.3. Scaling Speech Decoding With Unlabelled Data

Figure 3 shows that performance scales predictably with unlabelled data volume, following distinct patterns for differ-

---

*Due to the large computational cost of processing embeddings for MEG data in BrainBERT, we restrict pre-training of experiments in part C to approximately 30 hours and use only subject `001` of the downstream dataset for training and evaluation.

ent tasks and datasets. For speech detection on Armeni et al. (2022), we observe logarithmic scaling in log-space (log-log scaling), suggesting diminishing but continued returns with increased data. For other tasks, ROC AUC improves log-linearly, indicating robust scaling potential. Importantly, even our smallest pre-training dataset beats chance performance, while our largest (160 hours) continues to show gains without plateauing. Notably, we have scaled far beyond the data regime of prior surgical and non-surgical work and yet performance has continued to scale. Thus, our self-supervision approach may remain useful as the volume of open data in the field continues to rapidly increase.

Our results also reveal several new and notable phenomena. We scaled up the pre-training dataset by increasing the number of subjects. Since this led to consistent and almost monotonic improvements in downstream accuracy, our method is an exception to the consensus that pooling subjects worsens generalisation. As we pre-trained our model with a *different* dataset to those we fine-tuned on, our representation shows *cross-dataset generalisation*. This is surprising as the Armeni et al. (2022), Gwilliams et al. (2023), and our pre-training dataset all use different scanners entirely. Performing well across these datasets indicates that, together, our architecture and pretext tasks successfully generate representations that are generalisable across heterogeneous scanners. Finally, we note that our pre-training dataset contained no language data whatsoever yet still improved downstream accuracy on language tasks. Remarkably, this shows that unlabelled brain data collected from *any* task (including those that are not linguistic) can be used to improve speech decoding performance.

Since the results show improvements on both downstream tasks, this indicates that our pretext tasks are sufficiently generic to produce representations that work with multiple speech decoding tasks while still generalising well on each task individually. This is generally a challenging trade-off to manage. However, we notice that in both tasks, the base accuracy is higher and the improvement in ROC AUC is steeper for Armeni et al. (2022). This is likely to be because this dataset has more within-subject data. The weaker results for Gwilliams et al. (2023) may be a consequence of shorter intra-subject recordings, greater subject variation, and the lack of head casts in data collection. These observations support the findings of work such as Csaky et al. (2022).

### 4.4. Scaling Unlabelled Data Improves Generalisation to Novel Subjects

In neuroimaging, brain data is generally highly variable across participants, leading to difficulty transferring models to novel subjects (Csaky et al., 2022). Whilst we have shown generalisation *across* subjects, here, we investigate whether we can generalise to *novel* subjects—an even more

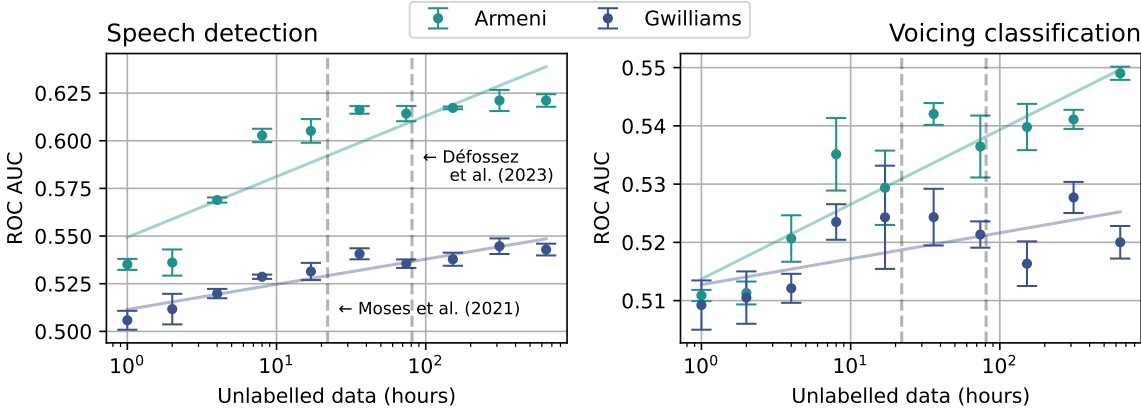

Figure 3: **Scaling unlabelled data improves generalisation.** We pre-train the model on increasing amounts of unlabelled data from Cam-CAN (Shafto et al., 2014; Taylor et al., 2017). The solid lines are linear fits and the dashed lines show the volume of data used in prior surgical (Moses et al., 2021) and non-invasive (Défossez et al., 2023) work. Unlike Table 2 (7) we do not apply Gaussian filtering to the predictions for simplicity. The best improvements are statistically significant.

difficult challenge. This is critical in order to widely deploy speech BCIs for new patients. In this experiment, we use Gwilliams et al. (2023) as our downstream dataset because of the large number of participants, holding out three subjects with which we evaluate novel subject generalisation.

Figure 4 shows that scaling up the amount of unlabelled data used in pre-training not only improves accuracy on subjects previously seen, but also demonstrates a positive log-linear trend in performance for novel subjects. This indicates that scaling our method is an encouraging direction for resolving the challenges of subject variance faced by prior work. As far as we are aware, this is the first result to demonstrate *novel* subject generalisation in speech decoding from MEG.

### 4.5. Aggregating Unlabelled MEG Datasets

Given the promising scaling results with single datasets, a natural question arises: can we achieve even better performance by combining multiple MEG datasets? This is particularly challenging since datasets often use different scanning hardware and experimental protocols. Thus, it has so far not been shown in MEG.

As a preliminary investigation, we combine two of the largest public MEG datasets: MOUS (Schoffelen et al., 2019) and Cam-CAN (Shafto et al., 2014; Taylor et al., 2017). In this section, we investigate how pre-training with these combined datasets affects downstream performance using the same experimental setup as Figure 3.

The results in Table 3 show, for the first time, that combining datasets can improve performance on downstream speech decoding tasks. It leads to better performance compared to pre-training on either dataset alone. It is surprising that

Table 3: **Aggregating unlabelled datasets outperforms single studies in speech detection.** For the first time with MEG, we show that unlabelled pre-training data from multiple studies with different hardware profiles can be aggregated while gaining the benefits of scaling. Combining data leads to a significant ($p < 0.05$) improvement.

| Pre-training Data | Hours | ROC AUC |
|---|---|---|
| CamCAN[1,2] | 159 | $.630_{\pm.0001}$ |
| MOUS[3] | 160 | $.614_{\pm.0004}$ |
| CamCAN[1,2] + MOUS[3] | 319 | $\mathbf{.638}_{\pm.0002}$ |

[1] Shafto et al. (2014) [2] Taylor et al. (2017)
[3] Schoffelen et al. (2019)

pre-training on Cam-CAN was better than pre-training on MOUS given that MOUS and the downstream dataset both used speech tasks and were acquired on the same MEG scanner. Cam-CAN, by contrast, did not use a speech task and was acquired on a different MEG scanner. We hypothesise that the better results for Cam-CAN are due to it being cleaner. During our experiments, we found that data quality, even among unlabelled data, can have a significant effect as artefacts in recordings disrupt learning.

While the combination of the two datasets includes far more hours of data than any prior work on deep learning with MEG, further work needs to be done to aggregate more datasets. Here, we were limited by compute budget and data availability. Increasing the number of datasets (e.g., by including EEG too) could further improve results. Just as increasing the number of subjects (rather than only within-subject data) improves novel subject generalisation, a larger

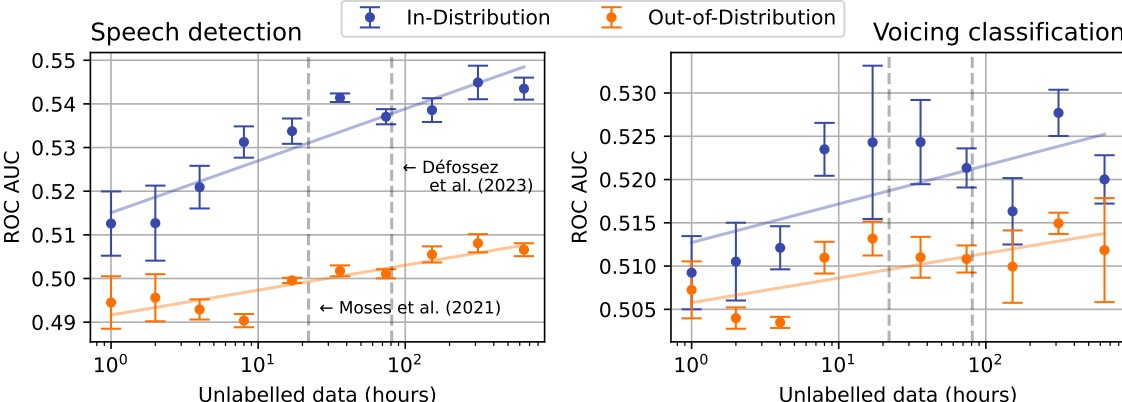

Figure 4: **Scaling unlabelled data improves novel subject generalisation.** We train a linear probe on Gwilliams et al. (2023). When *in-distribution*, we evaluate on held-out sessions; when *out-of-distribution*, we evaluate on held-out subjects. The lines represent the same as in Figure 3. The best improvements are statistically significant.

number of datasets may be key to scaling results when datasets are aggregated in pre-training.

### 4.6. Limitations

Although our results are significant in demonstrating a viable path forward to scale up speech BCIs, there remain a number of limitations to the present work. We focused here on two downstream tasks: speech detection and voicing classification. Ultimately, we would like to expand this work to predict full transcripts from brain recordings (i.e. *brain-to-text*). This has been achieved with surgical data (Moses et al., 2021; Willett et al., 2023) but not yet convincingly with non-invasive methods like MEG or EEG (Jo et al., 2024). Speech detection has played an important role in the development of full brain-to-text in a surgical context (Moses et al., 2021) and we hope may play a similar role for non-invasive methods. In future work, we would also like to expand classification to all English phonemes as a step towards full transcript decoding.

Additionally, while we have been able to demonstrate the utility of a few pretext tasks, we do not claim to have exhausted the full set of useful tasks. Rather, we conjecture that more useful pretext tasks remain to be found and believe a useful avenue of research will be into other input representations for brain recordings. For example, this paper did not make use of spatial features when the geometry of a scanner's sensor configuration is strongly correlated with the area of the brain from which the signal is derived. Another limitation is our emphasis on heard speech over other types of speech, such as attempted or imagined speech. We hypothesise that the same methods presented here will generalise to these other varieties of speech, though this has yet to be shown.

Perhaps the biggest limitation of the present work is that, while it surpasses the amount of data used in other studies, it remains to be seen how much speech decoding tasks can be improved by scaling up the number of datasets used in training. In sharing this work now, we believe that the current proof of concept will be sufficiently impactful to the field as we continue to actively scale up the datasets that we can leverage.

## 5. Conclusion

Speech decoding from the brain has been limited by the field's inability to scale up data to leverage deep learning. Prior methods have been unable to aggregate data across different datasets, labels, or subjects to scale up because of heterogeneity in recording hardware, experiment design, and participants. A handful of studies have shown weak signals towards alleviating these issues. But until now, no one has developed a general solution. We present a unified solution through data-efficient, self-supervised pretext tasks that overcome these fundamental scaling challenges. Our experiments demonstrate not just scaling with heterogeneous data, but generalisation across datasets, subjects, and tasks. They also show significant improvements of up to 27% compared to the prior state-of-the-art and even provide evidence of matching surgical decoding performance. Our method unlocks the potential of the bitter lesson, providing a general method to exploit more computation by using more data. We implore the research community to employ the vast quantities of data and compute available to realise this potential. If scale is all you need in speech decoding, then the bitter lesson may not be so bitter.

## Acknowledgements

We would like to thank Botos Csaba for many early insightful discussions which helped shaped the direction of this work. In alphabetical order, thanks also to Mats W.J. van Es for technical assistance with the OSL library, Yonatan Gideoni for advice on data splits, Minqi Jiang for an encouraging conversation on scaling unsupervised representation learning, Brian Liu for technical contributions which did not reach the final paper, and Miran Özdogan for reviewing a draft of this work. Finally, we thank the rest of the PNPL group for their continued and unwavering support.

The authors would like to acknowledge the use of the University of Oxford Advanced Research Computing (ARC) facility in carrying out this work. http://dx.doi.org/10.5281/zenodo.22558.

DJ is supported by an AWS Studentship from the EPSRC Centre for Doctoral Training in Autonomous Intelligent Machines and Systems (AIMS) (EP/S024050/1). GL is supported by an EPSRC Studentship. MW is supported by the Wellcome Trust (106183/Z/14/Z, 215573/Z/19/Z), the New Therapeutics in Alzheimer's Diseases (NTAD) study supported by UK MRC, the Dementia Platform UK (RG94383/RG89702) and the NIHR Oxford Health Biomedical Research Centre (NIHR203316). The views expressed are those of the author(s) and not necessarily those of the NIHR or the Department of Health and Social Care. OPJ is supported by the MRC (MR/X00757X/1), Royal Society (RG\R1\241267), NSF (2314493), NFRF (NFRFT-2022-00241), and SSHRC (895-2023-1022).

## Impact Statement

This work uses publicly available datasets from human studies (Armeni et al., 2022; Gwilliams et al., 2023; Shafto et al., 2014; Taylor et al., 2017; Schoffelen et al., 2019), each with their own ethical approvals and documentation available in their respective publications.

Neural speech decoding research has transformative potential for healthcare and assistive technology. Advances could help paralysed patients communicate freely and assist those with communication difficulties. By developing non-invasive methods, the field opens up the possibility of broader access to these technologies without the risks of surgical implants.

However, we acknowledge potential societal risks as this technology matures:

- Privacy and Data Protection: Brain signals contain highly sensitive personal information, raising concerns about data security and individual privacy.

- Consent and Misuse: Advanced decoding capabilities could enable unauthorized access to neural information, requiring robust safeguards against exploitation.

- Societal Impact: Widespread adoption could affect privacy norms around inner speech, while unequal access could exacerbate existing inequalities.

We focus specifically on decoding heard speech rather than inner speech, limiting potential misuse. Nevertheless, we recognize that advances in heard speech decoding contribute to the broader development of neural decoding technology. We encourage the research community to actively engage with these ethical considerations as the field progresses.

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

# A. Experiment Details

**Pre-training data.** We pre-train with non-overlapping sample windows from all subjects and sessions. We adjust the amount of unlabelled data used from Cam-CAN by increasing the number of subjects in the sequence 1, 2, 4, 8, 17, 36, 74, 152, 312, and 641, successively randomly selecting more subjects to include. Each seed uses a different set of subjects to reduce negative effects from outlier subjects.

**Labelled training data.** When training with Armeni et al. (2022), we hold out session 009 for validation and 010 for testing. Similarly, when fine-tuning with Gwilliams et al. (2023), we hold out task 1 from subjects 23, 24, 25, 26, and 27, using these sessions for evaluation only. As there is limited within-subject data in the latter dataset, we did not hold out a session from all subjects as before. For our novel subject experiments, we hold out subjects 1, 2, and 3 entirely and use the data for these subjects during evaluation. In Gwilliams et al. (2023), we note that they use four different tasks for each subject and their order is randomized between subjects. Both sessions for each task are repeats of the task. This means that while the recording itself is unseen, in this dataset, it is possible that held-out sessions use stimuli that may be shared.

**Adapting models for more sensors.** As BrainBERT is designed for single-electrode representations, for a fair comparison, to take into account all sensors, we ensured our linear classifier is applied to a concatenated embedding over all sensors. This is a large vector and leads to very computationally expensive training and is the reason we had to reduce the pre-training and downstream data in part C of Table 2. We similarly concatenate sets of 19 sensors when evaluating EEGPT.

**Statistical testing.** Significance was determined using one-sided $t$-tests with $p < 0.05$ as the threshold for significance.

# B. Hyperparameters

We conducted a search over hyperparameters of interest to optimise our self-supervised objectives and neural architecture. While these ablations indicated a theoretically ideal architectural configuration, in practice, we altered our final experimental architecture due to instabilities during training when data was scaled up. Our final architecture hyperparameters achieve a balance between the best values from our hyperparameter search and stable training. These values are detailed in Table 4.

Table 4: **Experimental hyperparameters.**

| Hyperparameter | Value |
| --- | --- |
| Window length (s) | 0.5 |
| $\rho$ (phase) | 0.5 |
| $\rho$ (amplitude) | 0.2 |
| $\{w_1, w_2, w_3\}$ | $\{1.0, 1.0, 1.0\}$ |
| $d_{\text{shared}}$ | 512 |
| $d_{\text{backbone}}$ | 512 |
| SEANet convolution channels | $(512, 512, 512, 512)$ |
| SEANet downsampling ratios | $(5, 5, 1)$ |
| FiLM conditioning dimension | 16 |
| Subject embedding dimension | 16 |
| Pre-training epochs | 200 |
| Optimizer | AdamW (Loshchilov & Hutter, 2019) |
| Learning rate | 0.000066 |
| Train ratio | 0.8 |
| Validation ratio | 0.1 |
| Test ratio | 0.1 |

**Why $\rho$?** The choice of the proportion of sensors to apply transformations to, $\rho = 0.5$ for phase shift prediction and $\rho = 0.2$ for amplitude prediction, were determined through a hyperparameter search. It is important to note that $\rho >= .5$ leads to the same effect as $1 - \rho$ for the complementary amplitude or phase shift. We conjecture that a smaller $\rho$ is optimal for amplitude scale prediction since this leads to representations that are especially strong at discriminating amplitude differences among

small groups of sensors. Perhaps this makes it easier to distinguish between neural responses from distinct parts of the brain such as the STG, which is associated with speech onset (Hamilton et al., 2018). In contrast, a larger $\rho$ for phase shift prediction could lead to representations that better discriminate neural synchrony information which is distributed across the brain rather than localised. As a result, a large proportion of the sensors in a MEG scanner should encode information about this feature.

## C. Compute Resources

All experiments were run on individual NVIDIA V100 and A100 GPUs with up to 40GiB of GPU memory on a system with up to 1TiB of RAM. Each pre-training run with the maximum amount of pre-training data took approximately 200 hours (8.3 days). Fine-tuning following pre-training took up to another 12 hours. We estimate that we used approximately 3000 hours of compute for the final experimental runs, including hyperparameter searches. In total, over the course of developing this work from idea to final paper, we used around 10,000 hours of GPU compute.

## D. Licences For Datasets And Code

The Armeni et al. (2022) dataset is distributed under CC-BY-4.0 while the Gwilliams et al. (2023) dataset is distributed under the CC0 1.0 Universal licence. The Schoffelen et al. (2019) dataset is distributed with a RU-DI-HD-1.0 licence from the Donders institute. The licence for the Cam-CAN (Shafto et al., 2014; Taylor et al., 2017) dataset is unknown. The SEANet code adapted from Défossez et al. (2022) is distributed under the MIT licence, and the OSL library, which we use for preprocessing, is under the BSD-3-Clause licence.

