# OpenReview forum: "The Brain's Bitter Lesson: Scaling Speech Decoding With Self-Supervised Learning"
_ICML.cc/2025/Conference — ICML 2025 poster_

### Official Review · Reviewer_RHbR · 2025-03-06

**Overall Recommendation:** 3

**Summary:**

This paper introduces a pre-training strategy for MEG recordings, which consists in neuroscientifically-grounded pretext tasks.
It shows scaling laws on two downstream tasks on two different datasets.

## update after rebuttal
Since the authors did not provide additional results or modifications along my suggestions, my score remains the same

**Claims And Evidence:**

Yes

**Essential References Not Discussed:**

None that come to mind

**Experimental Designs Or Analyses:**

The experimental design makes sense to me

**Methods And Evaluation Criteria:**

Yes. The authors follow the standard procedure of evaluating their pretrained model on downstream tasks, and provide comparisons with two baselines and two competing methods.

**Other Comments Or Suggestions:**

None

**Other Strengths And Weaknesses:**

Strengths
- Novelty: novel approach with the pretext tasks (differs from the usual self-supervised losses), and first to consider MEG
- Soundness: the experiments presented are sound and well executed (although not very exhaustive)
- The paper is well-written and easy to follow

Weaknesses:
- Missing results: I may have misunderstood the sentence "the backbone uses randomly initialised and frozen weights", but it looks to me as if the authors do not report anywhere two important points of comparison: (i) results obtained when finetuning the full model (not just a linear layer on top) on the labeled data of the downstream tasks, and (ii) the same but training the full model from scratch (no pretraining).
- The results are clean, but not particularly impressive: the downstream tasks chosen are rather easy tasks (binary classification). I appreciate that the authors wanted to display very clear scaling laws, and that has certainly been achieved, but it would be nice to also consider some more challenging tasks such as phoneme classification (even if results are poor). Similarly, the results on CamCAN+MOUS hardly outperform pretraining on CamCAN alone: although the gap might be statistically significant, I'm not sure how relevant it is; this section seems preliminary to me.
- Clarity: although the paper is well-written, there are some missing details. (i) After going back and forth several times, I could not clarify which task Table 2 reports results for: is it speech classification, voicing classification, or something else? Similarly for table 3. Please clarify this in the caption of the tables. (ii) I find it confusing that the plots in figures 3 and 4 look very similar, but in fig 3 the colors represent the datasets and in fig 4 they represent in-distribution vs out-of-distribution. Please change colors or find a different way of plotting

**Questions For Authors:**

“The latter dataset is particularly difficult to decode from as there is very little within-subject data and it did not enforce the use of head casts to immobilise participants” --> Why not use classification of global rotation of sensors as another pretext task (similar to random rotation augmentations used in computer vision)? It seems as if this could alleviate the aforementioned issue

**Relation To Broader Scientific Literature:**

In the neuroscientific literature, there has been a relatively limited amount of work on pre-training deep learning models compared to other fields, in part due to the difficulty to handle cross-dataset distribution shifts.
Existing approaches (e.g. BENDR, EEG2Rep, BIOT) typically involve self-supervised losses, where the input is masked or perturbed in some way and must be reconstructed, and only consider EEG recordings (rather than MEG).
This paper takes a novel approach which consists in using neuroscientifically-grounded pretext tasks, and is (to the best of my knowledge) the first to consider MEG recordings.

**Theoretical Claims:**

N/A

---

> ### Author Rebuttal · Authors · 2025-03-28
>
> Thank you for taking the time to provide a review. We are glad that you found the paper easy to follow and that our experiments were sound and well-executed. Please find below, our responses to your question and concerns:
>
> > authors do not report anywhere [...] results obtained when finetuning the full model
>
> Thanks for pointing this out. You’re right that we have not provided results when fine-tuning the full model. Here, we focused mainly on demonstrating that our self-supervised representation is highly generalizable. The simplest way to do this is through linear probing as it allows us to cleanly separate the contribution of self-supervision to the contribution of supervised learning. Fine-tuning the full model, would, however, be more thorough and likely provide better results. We prioritised demonstrating the SSL method over this given time constraints. We are working on collecting this result and others for posterity.
>
> > the downstream tasks chosen are rather easy tasks
>
> We appreciate this perspective, but want to clarify that these tasks are surprisingly challenging in MEG. The signal-to-noise ratio in non-invasive neural recordings is exceptionally poor. Recent work in [A] demonstrated that more complex decoding tasks (e.g. transcript decoding) perform no better than chance with current methods. Similarly, so far, no work that we are aware of has demonstrated strong results for speech detection or voicing classification on MEG which we believe highlights the difficulty of the challenge even here. State-of-the-art works in MEG speech decoding have also used similar tasks such as [B] and their segment identification task. Theirs is also less relevant for a speech BCI as it’s based on paired audio and brain data.
>
> You are right that ultimately the objective is to reach more complex tasks in order to achieve full speech BCIs. To this end, we are actively working on full phoneme and word classification as follow up work.
>
> > [...] which task Table 2 reports results for
>
> Thank you for noting this. It is for speech and we will clarify this in the captions for Table 2 and 3.
>
> > I find it confusing that the plots in figures 3 and 4 look very similar
>
> Yes, we agree that this is confusing. We will update figure 4 to use different colours. Thanks for bringing this to our attention.
>
> > Why not use classification of global rotation of sensors as another pretext task
>
> Indeed, as we discuss in the limitations section, we did not pursue any spatial pretext tasks. Your suggestion of a global sensor rotation task could be particularly useful in learning to account for head position changes. Thank you. Beyond movement compensation, phoneme perception also seems to have a strong spatial signature in brain activity [C, Fig. 4] and could further benefit from your suggestion. We are working on a small-scale experiment to test this.
>
> On a related note, some MEG datasets (which use scanners built by Electa) have MaxFilter [D] programs which can be applied to automatically compensate for head movements. However, this is not general enough for scaling as not all datasets will use these scanners.
>
> Thank you once again for highlighting some important points that have helped to improve our paper. Do you have any further questions or concerns?
>
> [A] Jo, H., Yang, Y., Han, J., Duan, Y., Xiong, H. and Lee, W.H., 2024. Are eeg-to-text models working?. arXiv preprint arXiv:2405.06459.
>
> [B] Défossez, A., Caucheteux, C., Rapin, J., Kabeli, O. and King, J.R., 2023. Decoding speech perception from non-invasive brain recordings. Nature Machine Intelligence, 5(10), pp.1097-1107.
>
> [C] Joan Orpella, Francesco Mantegna, M. Florencia Assaneo, David Poeppel; Decoding imagined speech reveals speech planning and production mechanisms; bioRxiv 2022.05.30.494046; doi: https://doi.org/10.1101/2022.05.30.494046
>
> [D] https://ohba-analysis.github.io/osl-docs/pages/docs/preprocessing-maxfilter.html

---

### Official Review · Reviewer_Bk17 · 2025-03-10

**Overall Recommendation:** 3

**Summary:**

This paper presents a unified solution through data-efficient, self-supervised pretext tasks to improve the speech detection and voicing Classification tasks. The experiments demonstrate significant gains from self-supervised pre-training. The method surpassed the baselines and is comparable to the model trained with surgical data. The data ablation (data size and data source) provides interesting insights to the community.

**Claims And Evidence:**

The claims made in the submission are supported by clear and convincing evidence.

**Essential References Not Discussed:**

The references look good.

**Experimental Designs Or Analyses:**

Experimental design is valid, and the analyses are good.

**Methods And Evaluation Criteria:**

They make sense.

**Other Comments Or Suggestions:**

N/A

**Other Strengths And Weaknesses:**

The strengths of this paper:

1) The paper is well organized and written.
2) The experiments are sufficient to validate the hypothesis.

The Weaknesses of the paper:
1) The designed pre-text tasks might be suitable for simple tasks only. The gain might be limited on transcript tasks.
2) Data scaling might be over-claimed as 1000h is a relatively small data size as compared to the speech recognition tasks.

**Questions For Authors:**

Several minor questions:

1. In Table 3, it shows that the cleaner data helps more on the finetuning tasks (CamCAN better than MOUS). The addition of MOUS data on CamCAN does not seem to improve a lot actually. Have the authors considered about data filtering method? Some data in MOUS might have negative effect on the pretraining tasks.

2. I am not familiar with surgical shown in Tabel 2. How about applying the proposed method for surgical as what was done for BrainBERT2+linear?

**Relation To Broader Scientific Literature:**

This is an incremental work by studying more pretext tasks and show its effectiveness on down-stream tasks.

**Theoretical Claims:**

This is not a theoretical paper.

---

> ### Author Rebuttal · Authors · 2025-03-27
>
> Thank you for your efforts in reviewing our work. We are glad that you found the paper well written and organized, and that the experiments validated our hypothesis. Below, we have provided our responses:
>
> > The designed pre-text tasks might be suitable for simple tasks only. The gain might be limited on transcript tasks.
>
> Yes, thank you for highlighting this consideration. It is an important tradeoff. If the pre-training task is overly specific, then it will likely be easier to learn but the representation’s utility will be limited for more general and complex downstream tasks. Ultimately, in the extreme, the most general pre-training task is next-step prediction. However, unlike some of the baselines we compare to (e.g. BrainBERT), we did not pursue this as the data scale in MEG is limited and, given most of the signal is noise, it is a very difficult pre-training task to learn well. Instead, we aimed for balance by designing pre-text tasks that are more specific than next-step prediction, through targeting useful features in brain activity, but general enough that they work well with downstream tasks related to neural speech decoding. This balance led us to achieve better downstream performance than comparable methods while generalising across multiple tasks, datasets, and subjects.
>
> We do share your interest in extending this work to full speech transcription. Jo et al. 2024 [A] demonstrate that current EEG/MEG approaches haven't exceeded chance for full speech decoding, highlighting the fundamental challenges in this space. Our approach specifically addresses prerequisites for more complex decoding by addressing generalisation and data efficiency problems. We're actively developing these extensions in follow-up work to further support our hypothesis that some of these pretext tasks will also enable more complex decoding.
>
> > Data scaling might be over-claimed as 1000h is a relatively small data size as compared to the speech recognition tasks.
>
> While this may seem relatively small in comparison to modalities such as audio now, data scaling should be considered within the context of MEG, which is the focus of our work. Here, we have scaled well beyond the volume of data used in prior MEG work e.g. Défossez et al. 2023 [B]. We are approaching the scale of early work in the adjacent field of deep learning speech recognition from audio e.g. Speechstew [C]. Thus, with these efforts we hope to similarly scale up MEG to progress speech recognition from brain activity in the near future. We will make this point clearer to ensure that we are not over-claiming. Thank you for noting this.
>
> > Have the authors considered about data filtering method?
>
> Thank you for the suggestion. So far, we have only explored detecting and rejecting corrupted channels through a variance-based threshold (known as autoreject in neuroimaging [D]). However, you are right that there are perhaps better ways to deal with data filtering as some artefacts can be global (across all channels) e.g. heart beats, muscle spikes, breathing, etc. One way we could attempt to address this in future work is to use signal-space projection (SSP) [E] if the network is not learning to ignore these types of artefacts.
>
> > How about applying the proposed method for surgical as what was done for BrainBERT2+linear?
>
> This is an interesting suggestion. In our work, we opted to apply BrainBERT+linear to non-invasive data as that is what we are mainly concerned with. While surgical data is out-of-scope for our current work, it is certainly of interest to the community and something we may explore in the future.
>
> Thank you again for your review. You have helped elucidate some critical points in our work. Do you have any further questions or concerns?
>
> [A] Jo, H., Yang, Y., Han, J., Duan, Y., Xiong, H. and Lee, W.H., 2024. Are eeg-to-text models working?. arXiv preprint arXiv:2405.06459.
>
> [B] Défossez, A., Caucheteux, C., Rapin, J., Kabeli, O. and King, J.R., 2023. Decoding speech perception from non-invasive brain recordings. Nature Machine Intelligence, 5(10), pp.1097-1107.
>
> [C] Chan, W., Park, D., Lee, C., Zhang, Y., Le, Q. and Norouzi, M., 2021. Speechstew: Simply mix all available speech recognition data to train one large neural network. arXiv preprint arXiv:2104.02133.
>
> [D] https://autoreject.github.io/stable/explanation.html
>
> [E] https://mne.tools/stable/auto_tutorials/preprocessing/50_artifact_correction_ssp.html

---

### Official Review · Reviewer_32uy · 2025-03-11

**Overall Recommendation:** 2

**Summary:**

This paper proposes a framework about how to leverage self-supervised learning (SSL) to improve the decoding of speech from brain activity. The authors propose a approach that utilizes large-scale unlabeled MEG data to train models, thereby addressing challenges posed by individual differences and dataset heterogeneity. The method demonstrates the generalization capabilities across multiple datasets.

**Claims And Evidence:**

No.

**Essential References Not Discussed:**

See Methods And Evaluation Criteria.

**Experimental Designs Or Analyses:**

See Methods And Evaluation Criteria.

**Methods And Evaluation Criteria:**

The model architecture used in this study does not differ fundamentally from those in prior work. The paper primarily focuses on unsupervised pretraining for MEG and speech task decoding. However, pretraining before downstream tasks is not a novel concept, and the lack of comprehensive experimental evaluation may undermine the claimed contributions of the proposed method.

The downstream tasks in this paper are focused on Speech Detection and Voicing Classification. However, significant progress has already been made in Brain-to-Text tasks by numerous studies [1][2][3]. I believe that the limited scope of downstream tasks somewhat diminishes the contribution of this work.

References

[1] Zheng H, Wang H, Jiang W, et al. Du-IN: Discrete units-guided mask modeling for decoding speech from Intracranial Neural signals[J]. Advances in Neural Information Processing Systems, 2024, 37: 79996-80033.

[2] Chen X, Wang R, Khalilian-Gourtani A, et al. A neural speech decoding framework leveraging deep learning and speech synthesis[J]. Nature Machine Intelligence, 2024, 6(4): 467-480.

[3] Défossez A, Caucheteux C, Rapin J, et al. Decoding speech perception from non-invasive brain recordings[J]. Nature Machine Intelligence, 2023, 5(10): 1097-1107.

**Other Comments Or Suggestions:**

No.

**Other Strengths And Weaknesses:**

The writing in this paper is straightforward and easy to follow, with clear descriptions of figures and tables. However, the lack of robust downstream task experiments and comprehensive baselines weakens the impact and contribution of this work.

I must point out that the methodology presented in this paper is overly simplistic, as the use of unlabeled data for pretraining and fine-tuning on downstream tasks is not a novel insight. It is recommended to emphasize the originality of the proposed method and its exceptional performance across a variety of downstream tasks.

**Questions For Authors:**

No.

**Relation To Broader Scientific Literature:**

See Theoretical Claims.

**Theoretical Claims:**

The methodological framework proposed in this paper is relatively straightforward, involving pretraining through the setup of three proxy tasks and fine-tuning specific modules. However, in terms of performance, it only marginally surpasses the fine-tuning performance of previously proposed general foundational models such as BIOT and BrainBERT. Additionally, it lacks comparisons with neural large-scale models like LaBraM [1] and EEGPT [2].

References

[1] Jiang W, Zhao L, Lu B. Large Brain Model for Learning Generic Representations with Tremendous EEG Data in BCI[C]//The Twelfth International Conference on Learning Representations.

[2] Wang G, Liu W, He Y, et al. Eegpt: Pretrained transformer for universal and reliable representation of eeg signals[J]. Advances in Neural Information Processing Systems, 2024, 37: 39249-39280.

---

> ### Author Rebuttal · Authors · 2025-03-28
>
> Thank you for taking the time to provide a detailed review. We are glad you found the paper easy to follow. Please find our responses below:
>
> > significant progress has already been made in Brain-to-Text tasks by numerous studies [1][2][3]
>
> While [1] and [2] are impressive, they address a different context and problem with invasive data and [3] suffers fundamental limitations to scaling.
>
> Specifically, [1] and [2] work on surgical data which is much easier to decode due to the better signal-to-noise ratio. We focus on non-invasive decoding as it avoids the risks and complexity of surgery in implanting BCIs. Additionally, [1] and [2] do not show scaling, do not work on novel subjects, and for [2], do not leverage pretraining. Thus, although they are important works, they are not relevant to our problem context.
>
> [3] is the only milestone work so far in non-invasive speech decoding from MEG. We discuss in lines 58-63 (column 2) that “they do not demonstrate generalisation to novel subjects and retrain their model for new datasets rather than being able to generalise across datasets or pool them. Their method is also unable to incorporate data without corresponding audio labels and so does not scale with other kinds of tasks.” As a result, they are fundamentally limited in scaling up data.
>
> Thank you for noting these papers as they are still important and interesting works. We will ensure to cite [1] and [2] in our revised PDF.
>
> > I believe that the limited scope of downstream tasks somewhat diminishes the contribution of this work.
>
> Our scope is similar to that of comparable leading work [3] who use a similar but ultimately less practical task to speech detection. Their speech segment identification task matches arbitrary 3-second segments of audio and brain and could be picking up on lots of unknown features (e.g. patterns of silence). Because it requires paired audio and brain data, it is unlikely to be a useful task for future BCIs. We go beyond a single task by also looking at voicing classification and in contrast to [3], speech and voicing classification are also directly relevant to speech BCIs for segmenting phrases and distinguishing phonemes for word decoding.
>
> While we agree that the ultimate goal is full phoneme, word, and sentence decoding, our work provides essential building blocks that non-invasive approaches have failed to achieve so far. [A] demonstrate that current EEG/MEG approaches haven't exceeded chance for full speech decoding. Our approach establishes prerequisite foundations by solving generalisation and data efficiency problems. We are developing more complex tasks in follow-up work.
>
> > it only marginally surpasses the fine-tuning performance of previously proposed general foundational models such as BIOT and BrainBERT
>
> Our improvements are 15-27% over BIOT and BrainBERT. They are also highly statistically significant (p<<.05). We would classify this as substantial as it is even sufficient to bring non-invasive decoding up to BrainBERT’s surgical decoding accuracy. The improvements are also larger than those quoted in BIOT over its own baselines.
>
> > lacks comparisons with neural large-scale models like LaBraM [1] and EEGPT [2]
>
> We have now added this comparison. Thank you. Our method outperforms EEGPT:
>
> | Method | AUROC |
> | :---- | :---- |
> | EEGPT | .602 \+/- .006 |
> | Ours | .705 \+/- .003 |
>
> As the largest EEGPT model supports up to 54 channels and our dataset uses 269 sensors, we concatenate the embeddings of consecutive chunks of 54 channels so that we can fairly take into account information from all sensors before linearly probing. We will add this result to Table 1.
>
> Thank you again for noting this. We have already cited LaBraM and will do the same with EEGPT.
>
> > the methodology presented in this paper is overly simplistic, as the use of unlabeled data for pretraining and fine-tuning on downstream tasks is not a novel insight.
>
> Our primary contribution isn't just applying pre-training and fine-tuning, but developing novel neuroscience-informed objectives (amplitude scaling, phase shifting, and band rejection) that take steps to address three fundamental challenges:
>
> 1) Novel- and cross-subject generalisation (historically a major blocker for MEG research [B])
> 2) Data-efficient and scalable self-supervised learning from heterogeneous datasets (no other work in MEG has done this); and
> 3) Generalisation across tasks and datasets
>
> In practice, our results also show a significant jump over baselines. We will make sure to emphasise this more in the revised PDF.
>
> Thank you again for your efforts in reviewing. You have helped us clarify several important aspects of our paper. Do you have any further concerns?
>
> [A] Jo, H., Yang, Y., Han, J., et al., 2024. Are eeg-to-text models working?. arXiv preprint arXiv:2405.06459.
>
> [B] Csaky, R., Van Es, M.W., Jones, O.P. et al., 2023. Group‐level brain decoding with deep learning. Human Brain Mapping, 44(17), pp.6105-6119.

---

### Official Review · Reviewer_aJaK · 2025-03-14

**Overall Recommendation:** 4

**Summary:**

Current speech decoders are generally trained individually per subject and only on task-specific data. The authors propose an MEG-specific self-supervised learning objective to build representations from a vast quantity of unlabeled MEG data from several subjects and tasks. They then built decoders that used these representations to detect phoneme voicing or the presence of speech. The authors claim that these decoders beat previous state-of-the-art self-supervised methods, with the advantage of generalizing to unseen subjects.

**Claims And Evidence:**

The authors were thorough in their evaluation, but I believe it's not always clear which task the decoder is being evaluated on. Most tables mention a single ROC AUC score, whereas Figures 3 and 4 mention separate scores for speech detection and voicing classification. I am assuming that all scores are for speech detection unless otherwise noted (based on "We evaluate our methods primarily on speech detection"), but clarification is needed.

**Essential References Not Discussed:**

I am not aware of any missing references.

**Experimental Designs Or Analyses:**

The evaluation method seemed sound -- in particular, it seemed the authors were careful not to contaminate the test set for within- or across-subject evaluations (except for the stimulus as noted in Appendix A, which I think is not a concern for these decoding tasks.)

**Methods And Evaluation Criteria:**

1. The individual terms of the self-supervised loss are well motivated, but I am somewhat confused by one aspect of amplitude scale prediction. Based on the choice of $\rho$, could the correct amplitude not be ambiguous? For example, suppose $\rho=0.5, A=2$. Then either of these could be true:

    A. $x_s^A = 2x_s$ for each sensor $s$ that _was_ randomly sampled

    B. equivalently, $x_s^A = 1/2 \cdot x_s$ for each sensor that was _not_ randomly sampled. So it could be that $A = 1/2$.

    Does this term strictly require $\rho < 0.5$ to have an unambiguous class? I also wonder if there's a boundary effect near 0.5, and this is why a smaller $\rho$ like 0.2 was chosen (as discussed at the end of Appendix B).

2. The two decoding tasks (speech detection & voicing classification) seem like a good place to start evaluating the quality of the SSL method. But as the authors mention in Sec. 4.6, existing BCIs can decode more useful features like phoneme (or even word) identity. Have the authors looked at decoding higher-level features like these? (I expect that phoneme error will at least be above chance given the phoneme voicing detection results.) Based on Appendix C, it looks like >90% of compute was dedicated to pre-training the network. Since that component is task-agnostic, I don't think adding another decoding task like phoneme classification would be too computationally difficult, but it could make the overall contribution much more substantial.

**Other Comments Or Suggestions:**

Minor, but it may be helpful to visually differentiate Figures 3 and 4 more -- they're easy to mix up at a glance. Changing the line colors in Fig. 4 would likely be sufficient, since currently, the same dark blue means different things: "Gwilliams" in Fig. 3 and OOD in Fig. 4.

**Other Strengths And Weaknesses:**

The paper is written well, and the ablations and comparisons to other models are done well.

**Questions For Authors:**

1. In Sec. 4.5, the authors write:
    > During our experiments, we found that data quality, even among unlabelled data, can have a significant effect as artefacts in recordings disrupt learning.

    Does this specifically refer to a difference in quality between Cam-CAN and MOUS? If so, how was data quality judged? What kinds of artefacts were found in MOUS but not in Cam-CAN?

**Relation To Broader Scientific Literature:**

The authors demonstrate that their self-supervised pre-training method outperforms existing methods for MEG by making use of existing publicly-available data. In particular, their non-invasive decoder shows comparable performance to invasive ones, and they show for the first time generalization to unseen subjects. They also show that scaling increases log-linearly, suggesting there is still room for improvement.

**Theoretical Claims:**

No theoretical claims were made in the paper.

---

> ### Author Rebuttal · Authors · 2025-03-27
>
> Thank you for taking the time to provide a review. We are glad you found the paper to be well-written and the experiments and comparisons to be sound. Please find below, our responses to your questions and concerns:
>
> >  Most tables mention a single ROC AUC score [...] I am assuming that all scores are for speech detection
>
> Yes, where it is not specified the score is for speech detection. Thanks for highlighting that this is unclear. We will add the specific score to the table captions where relevant.
>
> > could the correct amplitude not be ambiguous?
>
> You are absolutely correct. Thank you for identifying this and demonstrating it with an example. There is indeed an ambiguity here and we must restrict rho to be < 0.5 in order to have an unambiguous class. We conducted our ablation with values between 0 and 0.5 as we were aware of this, but as you have identified, we did not make this clear in the writing. We will add a note about this to the revised PDF. This should similarly apply in the phase shift task.
>
> The boundary effect hypothesis is very interesting. Could you please explain why you believe this influences the selection of a smaller rho?
>
> > [...] as the authors mention in Sec. 4.6, existing BCIs can decode more useful features like phoneme (or even word) identity
>
> So far, only existing *invasive* BCIs have been able to decode more complex features convincingly [A, B, C]. Attempts with non-invasive signals for sentence decoding have not yet produced results statistically significant beyond chance level, failing replication studies [D]. Nevertheless, as you point out, our voicing results do imply better-than-chance phoneme recognition results. In our preliminary experiments, while the results were better than chance, they were not much beyond that. We are actively working on improving the decoding of phonemes (focusing on acoustic features) and of words (focusing on semantic features) in a follow-up work.
>
> > it may be helpful to visually differentiate Figures 3 and 4
>
> Yes, we agree. Thank you for noting this. We will change the colours in Figure 4.
>
> > Does this specifically refer to a difference in quality between Cam-CAN and MOUS?
>
> Yes. We used a method to automatically detect corrupted channels and remove them using a variance-based threshold (known as autoreject in neuroimaging [E]). The percentage of corrupted channels in sessions from MOUS was higher than in Cam-CAN suggesting that the quality of the data was not as good in general. Better data filtering is likely to be an important future direction to continue getting improvements from additional datasets.
>
> Thank you again for your review. You have highlighted several important points which will help improve our paper. Do you have any further questions or concerns?
>
> [A] Moses, D.A., Metzger, S.L., Liu, J.R., Anumanchipalli, G.K., Makin, J.G., Sun, P.F., Chartier, J., Dougherty, M.E., Liu, P.M., Abrams, G.M. and Tu-Chan, A., 2021. Neuroprosthesis for decoding speech in a paralyzed person with anarthria. New England Journal of Medicine, 385(3), pp.217-227.
>
> [B] Willett, F.R., Kunz, E.M., Fan, C., Avansino, D.T., Wilson, G.H., Choi, E.Y., Kamdar, F., Glasser, M.F., Hochberg, L.R., Druckmann, S. and Shenoy, K.V., 2023. A high-performance speech neuroprosthesis. Nature, 620(7976), pp.1031-1036.
>
> [C] Card, N.S., Wairagkar, M., Iacobacci, C., Hou, X., Singer-Clark, T., Willett, F.R., Kunz, E.M., Fan, C., Vahdati Nia, M., Deo, D.R. and Srinivasan, A., 2024. An accurate and rapidly calibrating speech neuroprosthesis. New England Journal of Medicine, 391(7), pp.609-618.
>
> [D] Jo, H., Yang, Y., Han, J., Duan, Y., Xiong, H. and Lee, W.H., 2024. Are eeg-to-text models working?. arXiv preprint arXiv:2405.06459.
>
> [E] https://autoreject.github.io/stable/explanation.html

---

> > ### Comment · Reviewer_aJaK · 2025-04-04
> >
> > Thank you for the thorough responses.
> >
> > > The boundary effect hypothesis is very interesting. Could you please explain why you believe this influences the selection of a smaller rho?
> >
> > Whatever my reasoning was (I don't exactly remember), I don't agree with the notion now.

---

### Decision · Program_Chairs · 2025-05-01

**Decision:**

Accept (poster)

**Comment:**

This paper presents a novel contribution to the field of machine learning, introducing an innovative self-supervised learning (SSL) approach for neural signals collected using magnetoencephalography (MEG). The proposed method demonstrates improved decoding performance across participants, which is essential for developing generic models that can be transferred effectively across individuals.

The reviewers acknowledge the novelty of this approach, particularly in its use of unique pretext tasks that deviate from traditional self-supervised losses. The execution of these tasks is deemed sound and well-implemented. While all reviews are generally supportive of the paper, one reviewer suggested extending the comparison with recent EEG models, which could provide additional insights. The authors have addressed this request during the rebuttal period.

Based on the overall positive feedback from the reviewers, I endorse this paper for publication at ICML.